# Precise Water Leak Detection Using Machine Learning and Real-Time Sensor Data

**João Alves Coelho [1,†], André Glória [1,2,*,†]**  **and Pedro Sebastião [1,2]**

[1] Department of Information Science and Technology, ISCTE—Instituto Universitário de Lisboa, 1649-026 Lisbon, Portugal; jmjac@iscte.pt (J.A.C.); pedro.sebastiao@iscte-iul.pt (P.S.)
[2] Instituto de Telecomunições, 1049-001 Lisbon, Portugal
* Correspondence: afxga@iscte-iul.pt
† These authors contributed equally to this work.

**Abstract:** Water is a crucial natural resource, and it is widely mishandled, with an estimated one third of world water utilities having loss of water of around 40% due to leakage. This paper presents a proposal for a system based on a wireless sensor network designed to monitor water distribution systems, such as irrigation systems, which, with the help of an autonomous learning algorithm, allows for precise location of water leaks. The complete system architecture is detailed, including hardware, communication, and data analysis. A study to discover the best machine learning algorithm between random forest, decision trees, neural networks, and Support Vector Machine (SVM) to fit leak detection is presented, including the methodology, training, and validation as well as the obtained results. Finally, the developed system is validated in a real-case implementation that shows that it is able to detect leaks with a 75% accuracy.

**Keywords:** internet of things; green tech; water management; machine learning; sustainability; water leaks; efficiency

## 1. Introduction

Nowadays, technology takes a major role in our lives and we are always searching for better solutions to solve everyday problems, mainly to improve and ease repetitive tasks in an autonomous way with less effort. This evolution in technology reached new potentials with the increase of devices being used every day, allowing for the concept of Internet of Things (IoT) to arise and be part of the proliferation of smart technologies and cities [1].

IoT is the evolution of the current Internet [2] and consists of a network of devices that are capable of collecting and controlling data from the physical world, with these devices being capable of perceiving, computing, executing, and communicating between users and things [3]. These networks of devices are usually associated with the concept of Wireless Sensor Networks (WSN) and multiple small nodes that communicate among them using wireless links, covering large areas and collecting data in real time. These enable control over the environment and an interaction with the users, with IoT and WSN being reliable features on the development of monitoring and control systems [4].

IoT systems can be applied in multiple scenarios, as they are composed to retrieve information about the operation, conditions, and performance of any task or environment that can be remotely controlled. Therefore, in the last years, several sectors have adopted these systems, for example, in health, transport, retail, buildings, and agriculture [5]. This last one is a major market for IoT not only to improve the performance of agricultural fields, with precise monitoring, but also due to the fact that waste of natural resources, mainly water, needs to be managed. Irrigation systems account for 70% of the world's fresh water used annually, and 30% of that water is potentially being wasted due

to environmental issues or lack of monitoring but mainly due to leaks in pipes that supply the water along the fields [6,7].

Water management depends on how well the use of water is maximized and the water losses are minimized [8]. This type of management requires precise analysis which may be too complex for humans to perform correctly, and with the increase in human population, sustainability and efficient water usage assume important roles. As water is a scarce resource, detecting problems with the supply and distribution of water as fast as possible can be achieved throughout a sensor network, leading to minimal to no waste in activities such as agricultural irrigation.

Typical solutions for leak detection in pipes include shutting down the water supply system and using acoustic devices to check if the sound can reach the end of the pipe without losing strength, meaning that the pipes do not have any leakage points. These kits, available on the market and widely used by plumbers and inspectors, are portable devices capable of detecting flaws in pipes with sound, such as in [9,10]. This solution not only requires additional manpower to be performed but also implies a break in normal activities of the system, meaning that no water can be used, leading to a potential reduction of performance in the irrigated field, making this type of analysis a maintenance checkup that occurs periodically instead of a real-time analysis. Another method commonly used is a visual inspection not only on the pipes themselves when they are exposed above ground but also on areas that have signs of flooding resulting from a burst pipe underneath. As in the previous method, this involves additional manpower and cannot be carried out on a real-time basis.

To improve the efficiency of water distribution and to reduce waste related to leaks, a sensor network can be used to execute those analyses with precision and in an autonomous way. Some research and development can be found in both the academic and industrial worlds, ranging from expansive fixed systems that use ultrasonic clamps to analyze real-time water flow, such as [11], to low-cost solutions found in the academic world, such as [12–16].

To optimize the daily management and control of possible leaks, this paper aims to create a system that can control and monitor water leaks through a system that collects data using a low-cost sensor network in order to evaluate possible leak points in the water system. The data obtained from the sensors is stored and treated by a Machine Learning (ML) algorithm that will allow the user to be notified if the water distribution system starts to leak, of the potential size, and of the location of those leaks. All interactions and notifications with the user are done through a mobile app, where they can also act upon and take care of the water leak. This work proposes a new system to monitor and control the water flow through a water distribution pipeline, having in mind the reduction of water wasted and a monetary saving for the final user. This paper not only contributes a new way to monitor leaks with a low-cost solution that includes a complete ready-to-use system but lso provides a detailed study on how machine learning can be used alongside sensor data to detect in real-time leaks in pipes based on that data as well as a comprehensive comparison between several algorithms in order to discover which best fits this problems.

The remainder of the paper is organized as follows: Section 2 describes the current research status on leak detection using IoT and ML, with some significant research being analyzed and compared with our approach. A detailed description of our system architecture, including hardware, for data collection, communications, data analysis, and visualization are described in Section 3. Section 4 presents our approach on the machine learning training, with the methodology, the creation of a training and testing dataset, and the results of the training methodology. Section 5 details the experimental setup to validate our system methodology, while Section 6 presents the obtained results and Section 7 presents the discussion of those results. Finally, the conclusions are outlined in Section 8.

## 2. Related Work

Water is a crucial natural resource and it is widely mishandled, with an estimated one third of world water utilities having a loss of water of around 40% due to leakage [17]. Traditionally, pipeline leakage detection requires periodical inspection with human involvement, which makes it slow and

inefficient for timely and fast detection. This leads to an increase in the development of methods to detect, locate, and fix leaks, and it is something we can see in numerous different projects.

As previously said, the market already offers solutions for professional use, mainly with portable devices that are used in maintenance situations and that require human assistance. Those devices are often composed of ultrasonic or acoustic sensors that detect leaks in empty pipes using sound, such as in [9–11]. The main disadvantage of those systems, apart from their expansive cost and requirement of manpower, is the lack of real-time analysis, which can only detect that a leak exists but is unable to detect when it occurs.

In the academic area, this field presents several papers, mainly without the use of ML techniques. In [12], the authors created a system to inform and control water usage in households and buildings. The system that is installed directly on pipes uses a water pulse meter to measure the intensity and continuity of water flow along the plumbing system in order to detect water leaks. The disadvantages of this system are that it only detected leaks if a sudden decrease in the water flow occured and that it cannot identify where the leakage occurred, only that it occurred. Also, the system was not tested on a real environment, so the accuracy of the concept is not available.

The authors of [13] also developed a WSN with thermal sensors to detect water leaks in pipes. In this case, the system detects temperature changes of the soil above the leakage, allowing the user to be warned of a possible leak location. The authors presented detection of a leak when the soil changes by 0.5 °C but also stated a major flaw in their system, since environmental conditions can deeply affect the results. Besides that, the system also lies on the assumption that the soil is composed of sand, thus not being able to work in concrete or brick.

To develop a smart water meter with leak-detection capabilities, the author of [14] used a set of simple sensors, mainly flow meters, to gather information about water flow in pipes, sending that information to a cloud system using LoRaWAN. To detect the presence of leaks, a simple comparison of adjutant sensors is performed, classifying a possible leak based on the drop of flow between those two points. Although the system is well presented and the methodology followed is similar to the one presented in this paper, our solution benefits from the fact that we use machine learning to detect leaks, allowing for a more detailed analysis. Also, our approach differs with the use of Narrow-Band IoT (NB-IoT) to send data to the cloud, as LoRaWAN gateways are still scarce, as is pointed by the authors. NB-IoT allows for more capability to install a system anywhere in the world.

When focusing on ML techniques to detect water leakage, fewer papers were found. In [15], a system of NB-IoT using ultrasonic sensors to detect, with precision, the location of leaks in the pipelines is presented. The system uses the Doppler effect from ultrasonic sensors to detect minimal change in water passing by the pipes and, when detected, informs the user where the leak happened. This data analysis is done using Support Vector Machine (SVM), with a 92% accuracy. Although the authors achieve great results, it is important to state that the system was trained and tested using only 200 samples and retrieved in a controlled environment, a situation that can justify the high accuracy of the system and does not prove that the system works on a real-case scenario. Besides that, this system requires expensive sensors which creates a system that it is not accessible to all users.

The authors of [16] used data from smart meters located alongside water distribution pipelines to train a set of ML algorithms in order to detect leaks. The methodology followed was training the algorithms on different pipes for one day, with some pipes not showing any leakage, and testing the trained models on a different day. The conducted training encountered 21 leaks in 8 pipes, which in our opinion is considered a small dataset, and during testing, leaks occurred 25 times, being detected by the ML models as follows: random forest—75–96%; decision trees—72–94%; and k-nearest neighbors (KNN)—76–92%. Despite presenting great results, it is still possible to conclude that the range of accuracy among the various models is wide, mainly due to the small dataset used. Besides that, the study is important to help us choose the algorithms to test in our system, mainly random forest, that shows better performance.

We can see that there are already various types of WSN solutions for water leak control and detection, but all have some limitations that reduce its efficiency in precisely detecting leaks. The main problem encountered with the existing solution is price or lack of intelligent systems to detect leaks, with only the values of two sensors being compared to check for pressure drop. Some solutions already present an integration of machine learning techniques, but it is possible to conclude that there is still a great deal of work to be done in this area, as studies ate still preformed using small datasets and therefore the results can be misleading.

Our solution aims to solve some of these issues. By using low-cost hardware and an easy-to-implement system, our solution is widely available for everyone, even farmers with low budgets. Also, using NB-IoT as an approach to send data to the cloud is beneficial, as the technology is starting to be the choice for this type of project. Finally, the integration of ML to help detect leaks is the major contribution and innovation of our system, with an additional contribution being the comprehensive study and testing of multiple algorithms to understand which best fits this type of solution.

## 3. Materials and Methods

To achieve the main objective of detecting leaks and their location in pipes in real time using machine learning, there was a need to design a control and monitoring system to be applied in water distribution pipelines in the public and private domains. This system intends to use multiple sensors with real-time data collection, mainly water flow parameters to improve the efficiency and the early detection of leaks, with the support of machine learning techniques. Figure 1 shows a possible implementation of the system in a drip irrigation system.

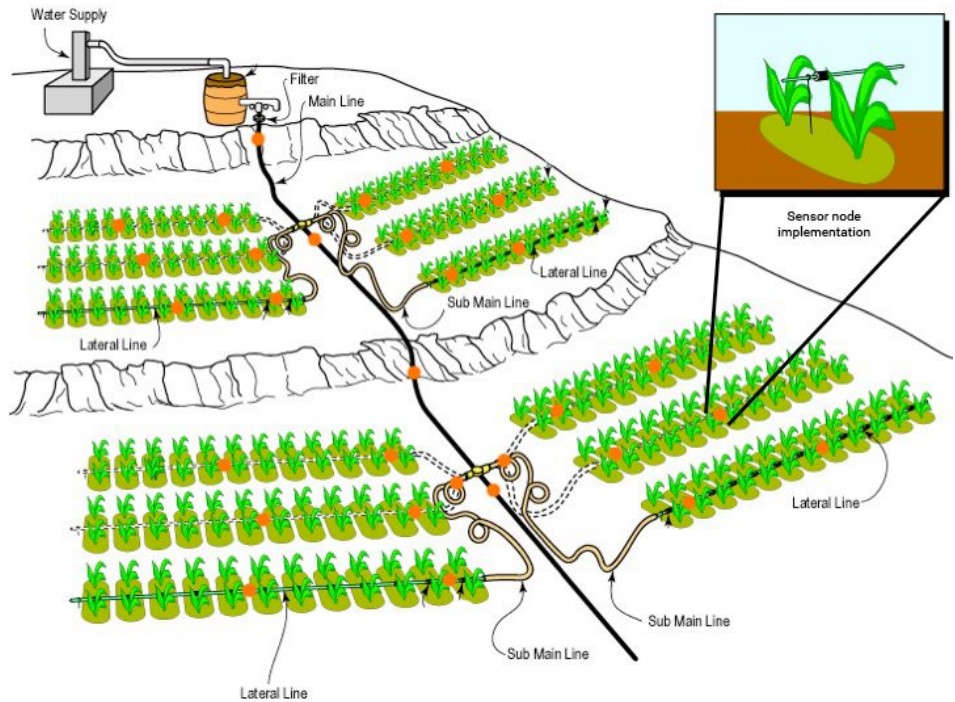

**Figure 1.** System implementation.

As can be seen in Figure 2, the system is divided in multiple modules, with both software and hardware parts, each with its individual purpose, from the hardware nodes, for data collection; the server, for data processing; and finally the user.

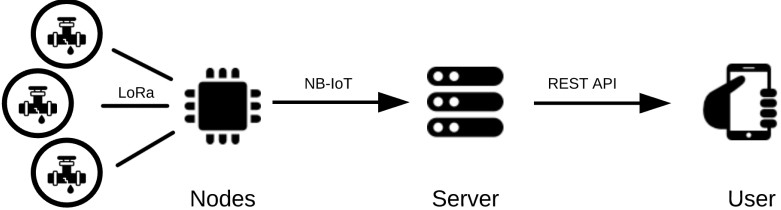

**Figure 2.** System architecture.

The system consists of various water flow measuring sensors, spread throughout the water distribution pipeline to collect information as water flows by them. This information is then sent to the aggregation node, the main hub of our system, responsible for communication with the server, transmitting the information collected from the sensors. In the server, the information is stored and is also run through the ML algorithm to be studied and interpreted. From here, depending on the analysis done by the algorithm, the information is shown to the user as being all normal within the system or will alert the user for potential locations of water leaks. This logic can be better comprehended in Figure 3.

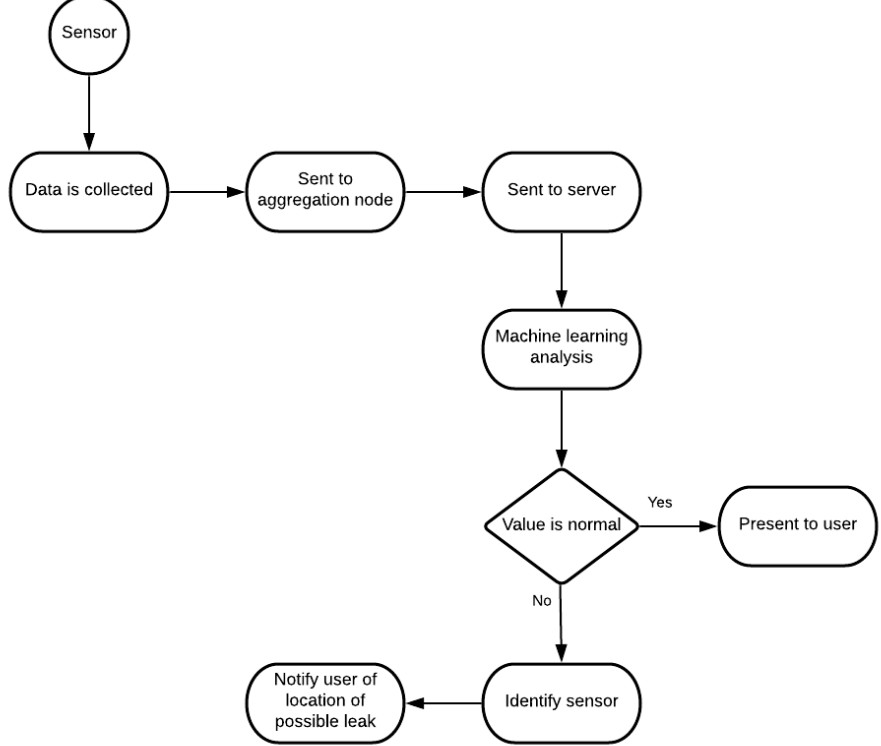

**Figure 3.** System logic.

### 3.1. Data Collection

The developed system consists of a network of sensors, as shown in Figure 2, that conjointly create a typical WSN, capable of retrieving data from the sensors and of sending them to a cloud server.

Each of the nodes was created to get maximum efficiency, low cost, and low power, with the best microcontroller, communication protocols, and supplementary hardware being used. For that, the ESP32 microcontroller was used as the basis of each node due to its dual core chip which includes Wi-Fi and Bluetooth Low-Energy (BLE), 32 general-purpose input/output (GPIO) ports that can be assigned different programming functions, 12 analog ports, and low power capabilities, with the

ability to enter into a deep sleep state. [6,18]. Another core feature and hardware that the nodes share is the ability to communicate via a LoRa peer-to-peer network using the RFM95W LoRa transceiver, that grants a long-range spectrum communication and immunity to high frequency, all while reducing power consumption. It supports high performance Frequency-shift keying (FSK) modes and delivers exceptional noise, selectivity, and receiver linearity for a much lower current consumption than other devices [19].

The specification for each node and additional hardware are explained in the following sections.

### 3.1.1. Aggregation Node

The aggregation node is the central node of the network and the one responsible for keeping the network connected. The aggregation node does not collect any data; it just sends the information it receives to the server. In our system, it works as a bridge between the sensors that gather the data and the server that stores the data.

In Figure 4, is possible to see how the aggregation node of our system is built. As said, the aggregation node receives messages from the other nodes via LoRa, and for the server communication, Message Queue Telemetry Transport (MQTT) is used via an NB-IoT connection, using the SIM7000E module. For that, the node cannot be connected to a battery, since it needs to always be listening for new messages, being connected directly to the electrical grid using a 5 V power supply.

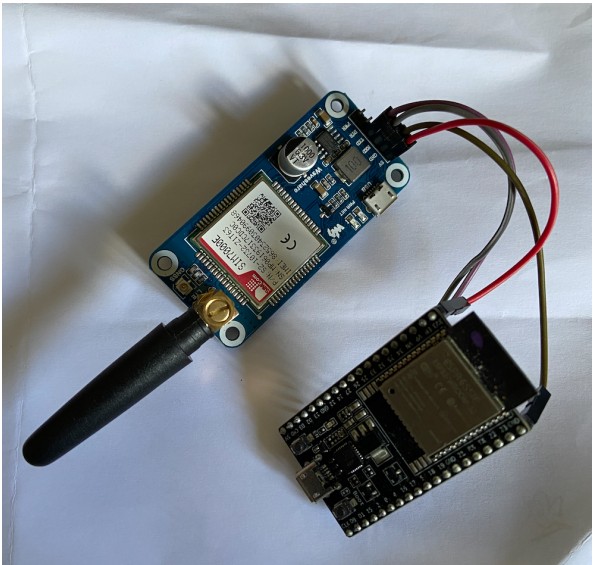

**Figure 4.** Aggregation node.

### 3.1.2. Sensor Node

The sensor node is the simplest and the lowest level of our WSN, with their sole purpose being to collect data from the attached sensors and to send it to the aggregation node. In order to perform these tasks, the sensor node needs a microcontroller and a set of sensors that can be permuted from node to node depending on the different needs for the solution.

In Figure 5 the sensor node is represented. The sensor nodes consist of, as mentioned before, an ESP32 microcontroller, a RFM95W module, and an array of sensors suited for retrieving information and sending it to the aggregation node. To transmit the information collected from the sensors to the aggregation node, it uses a LoRa connection through the RFM95W module.

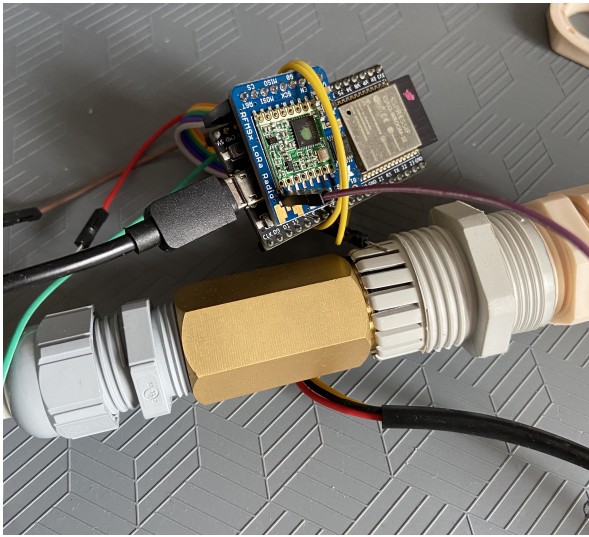

**Figure 5.** Sensor node.

With the only goal of measuring water flow in the pipelines, only one type of sensor is needed, the YF-B2 water flow sensor [20]. This sensor consists of a water rotor and a hall-effect sensor. When water flows through it, the rotor rolls, and its speed changes with the different rate of flow.

Since the node only needs to be awakened when water flows by, it can be powered with a battery, which allows for a completely wireless node and a longer lifetime. Table 1 shows the node power consumption on each set of its life cycle.

**Table 1.** Sensor node power consumption.

| Node Status | Power Consumption $V_{IN} = 3$ V |
|---|---|
| Transmitting | 80 mA |
| Collecting data | 30 mA |
| Deep Sleep | 100 μA |

Since the sensor node is in deep sleep mode when no water is distributed, which in an irrigation system is almost 80% of the day, and when irrigation starts, a sensor reading of 5 min each is made, being in deep-sleep in between samples during the irrigation period. As such, the node can be powered by a CR123 3 V battery for long periods of time. Table 2 presents the expected lifecycle of the batteries depending on the irrigation period per day.

**Table 2.** Sensor node average lifetime.

| Irrigation Period | Battery Lifecycle |
|---|---|
| 10 min | ±545 days |
| 30 min | ±500 days |
| 1 h | ±460 days |
| 3 h | ±335 days |
| 6 h | ±238 days |

To improve the lifetime of the sensor, small solar panels can be installed to charge a LiPo battery instead of the CR123, making the system autonomous.

*3.2. Communications*

Communication is the most crucial part of a WSN as it is responsible for transmitting the data collected from the sensors and to keep the network connected and to assure that the system performs the tasks specified.

The projected system, composed of two sets of nodes, requires two types of communication, one for communication between nodes and the aggregation node and the other for communication between the aggregation node and the server.

### 3.2.1. Node-To-Node Communication

Node-to-node communication is how information travels inside the WSN. Since nodes can be far from each other and are powered by batteries, a communication protocol capable of long-range transmission with low power consumption is essential.

With this in mind and in order to have all the nodes in the network connected, LoRa was implemented using a RMF95W module. LoRa is a physical layer that uses Chirp spread spectrum (CSS) modulation, with the same low power characteristics of the FSK modulation (which is part of many other communication protocols) but with an increased range [6,21]. It uses a star topology, which allows for lower power consumption and a reduction in the network complexity and capacity.

To implement a LoRa connection in the node, the RadioHead library was used [22], which enables node addressing, with unique addresses being set for each node, guaranteeing differentiation in the network. With the RadioHead library, the nodes in the network can communicate in two different ways: by sending a broadcasting message to every node with the destination ID included in the message and by directly messaging the node by using its destination address.

### 3.2.2. Node-To-Server Communication

Node-to-server communication is how information containing the data gathered from the sensors leaves the network. To connect with a server outside of the network, some kind of internet connection is required and this is where NB-IoT was used.

NB-IoT is a standard-based low-power wide-area network (LPWAN) system, developed to allow devices to connect through mobile phone signals [23]. It uses devices with low data rates that take advantage of the cellular network to communicate between each other but with the convenience of consuming less energy. The main focus of NB-IoT is to improve IoT solutions coverage with low-cost devices, a battery life of up to 10 years, and high connectivity.

NB-IoT was chosen over LoRaWAN, a variation of the LoRa protocol that uses public gateways that can cover thousands of squared meters to receive messages from nodes and to send them directly to the user server through an API without costs due to the low coverage that LoRaWAN still has worldwide. With this approach, implementation does not require the installation of a LoRaWAN gateway, for which the prices start at 300 € and require installation in strategic and high places to allow for good coverage and a good Internet connection, which are hard to find in remote locations such as agricultural fields. With an aggregation node using NB-IoT installed with the system, coverage is secured, since almost everywhere, a cellular connectivity is available, with the SIM7000E allowing a fallback to 2G when no NB-IoT connection is available. With the cost of this node being only around 30 €, the cost to install a LoRaWAN gateway is a fraction. Besides that, since the bandwidth needed for transmitting the data is very low, only 5 Mb per node per month when transmitting every hour, the associated costs for these transmissions are less than 0.50 € per month using IoT data services such as The Things Mobile.

In order to always maintain a connection between the network and the server, the MQTT protocol (available online: accessed on 15 March 2012) was selected based on its architecture that fits IoT projects and for its lower power consumption when facing typical HTTP requests.

MQTT is a communication protocol built on top of Transmission Control Protocol (TCP) protocol, with a low complexity and with the main goal of connecting devices to embedded networks. It is a many-to-many type of communication, and it consists of three components: publisher, broker, and subscriber [24]. The publisher, to communicate via MQTT, sends a message with two components: a topic and a message. The broker receives the message and distributes it between multiple devices. These devices register as subscribers for the specific topics of a publisher in order to gather data [25].

In this publish/subscribe model, the subscriber can only receive messages from the topic to which they have subscribed. This is ensured by the broker that also makes the different topics available. Contrary to HTTP, with this type of publish/subscribe model, it is possible for several publishers to communicate at the same time, so one node does not need to end its connection with the server for others to make a new connection [26]. Since it provides a simple implementation and provides routing for low-power, low-cost, and low-memory devices in low bandwidth networks [27], it is one of the most suitable connection protocols for IoT.

For this, the PubSubClient [28] library was used to implement the MQTT on the aggregation node.

### 3.3. Data Analysis

For the system to be able to collect and analyze in real time the data that is sent by the nodes, a set of scripts was developed. The proposed goal of the pretend system is to identify in real time the locations of possible water leaks, and for that, machine learning was used.

The first step for the system to be able to predict the presence and location of leaks is to receive information from the sensors and to preprocess them, including saving the values and corresponding timestamps in a database for future analysis and to feed information to the mobile application. The preprocessing consists of a set of prevalidations and calculations, such as guarantees that the values are within the expected range, calculated averages, and fluctuations between sensors.

For that, a Python script was developed using the Paho Python MQTT library [29] to be able to receive the values from the network.

Afterwards, the scripts put the received values as inputs in a machine learning algorithm to predict the final output and to warn the user if needed. The result from the ML analysis indicates whether the pipeline under study presents leaks and the section where the leak is located based on the location of the sensors. Although the precise location cannot be presented, it reduces the search area, as sensors are not far from each other, improving the efficiency of the system, as the pipe does not need to be completely checked, only the length between those sensors.

This was implemented in the same script, and the algorithm chosen for this task was previously trained and fitted as described in the next section.

### 3.4. Data Visualization

To provide the user with a way to see the data collected from the sensors, an Android mobile application was developed from scratch. The objective of the mobile application was to give the user a dashboard of the water distribution system, where it is possible to check all the values from the sensors in real time, everywhere.

Regarding the values retrieved from the sensor nodes, the application gives the user the opportunity to check in real time not only the latest values but also the current conditions of the corresponding pipes, as evaluated by the machine learning algorithms as well as historical data and conditions. For the real-time system, the Paho Java MQTT library [29] was used to subscribe to the topic to which the network publishes information.

If the user does not check the application on a regular basis, it will warn the user via a notification if some problem is encountered by the system.

In order for the user to have a visual interaction with the system and to be able to perform the described functionalities, the developed Android application is composed of a set of screens, as presented in Figure 6.

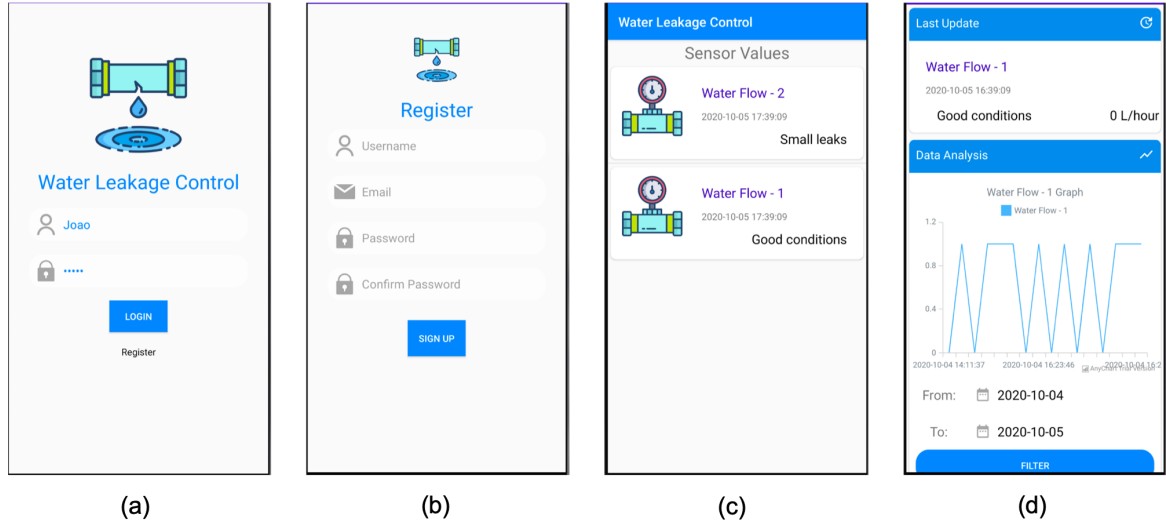

**Figure 6.** Android screens: (**a**) login, (**b**) register, (**c**) dashboard, and (**d**) sensor detail.

## 4. Machine Learning Study

As previously said, the central part of the system is the machine learning algorithm, which is used to predict water leaks and to inform the user of leak location. To be able to do this, the algorithm must first be trained to understand the system and the environment, so that, when a water leak occurs, the system may know what is happening and what to do.

One of the goals of this paper and one of the major author contributions, was to understand what is the best algorithm to use not only in our system but in other situations with similar data. For that, five different classification algorithms were tested to check which has the best accuracy.

For that, the following algorithms were tested:

1. Support Vector Machine (SVM) —used mostly for classification, it classifies the data by building *n* dimensions between two classes and by finding an optimal hyperplane to categorize the data, using the distance between the neighboring points and differentiating between the classes with minimum error margin [30]. In a simpler explanation, given training data, the algorithm outputs the best hyperplane that classifies new examples [31].

2. Decision Trees (DT)—through a hierarchical partition of training data, a certain feature is used to split the data, with this split being done iteratively until the leaf node contains a number of records that can be used to classify the data [32,33]. However, as described in [34], this algorithm faces some limitations since a small change in the training dataset can lead to a substantial change in the tree, making it harder to predict the next values with precision.

3. Random Forest (RF)—best applied for classification problems, this integrates the process of aggregation bagging and DT by choosing a subset of features from individual nodes of the tree, by avoiding correlation on the bootstrapped set [33], and by working with an assortment of trees in which each tree gives a classification.

4. Neural Networks (NN)—it functions in the same way as the human nervous system, using neurons in various layers in a biosystem-like shape. Each of the neurons analyzes parts of the input and sends the information to the next layer and neurons continuously, until it is able to reach a valid output [32]. It is ideally used in nonlinear and complex problems which requires large computational power and has some disadvantages when working with IoT systems due to low complex and low power devices [33]. In this case, Multilayer Perceptron (MLP), a variation of the NN algorithm which consists of multiple neurons organized into layers [35], was used.

5. XGBoost—with a similar model to DT, the objective of this algorithm is, as the name suggests, boosting the performance of the model. It creates a sequence of models, and rather than training all individually, it models in succession, so that the new models attempt to correct the mistakes

of the previous ones [36]. The first model is built on an original dataset, with the second model improving the first model, the third model improving the second, and so on. The models are added sequentially until no further improvements can be made.

### 4.1. Dataset Creation

In order to train the algorithm, first, it was necessary to create a dataset with similar data to the one that will be used in the future, containing all possible outputs. As if an output never happens in the dataset, the algorithm will not be able to predict it. In the intended system, the ML algorithm will only be used for analyzing a specific scenario: water leaks and its location.

For this scenario and in order to create the needed dataset, a small water distribution system was created in which our system was implemented, as shown in Figure 7.

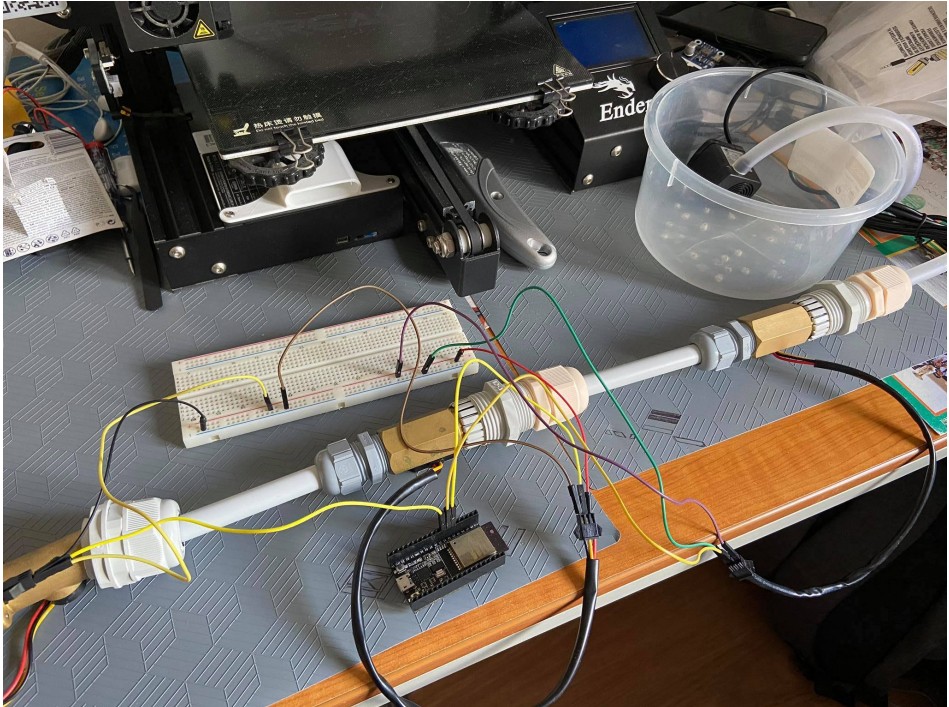

**Figure 7.** System prototype.

The system collected data every 5 min, about 12 data samples an hour, with all samples being recorded with a value collected and the corresponding pipe able to classify the collected information for all possible outputs.

To achieve these outputs, in the test scenario, different holes were carved in the pipes, with size variances and with pipes with more holes than others, to simulate a rupture in the system and to match the desired output. Table 3 shows the desired outputs and the corresponding situation in the system.

**Table 3.** Intended outputs.

| Output | Description |
|--------|-------------|
| 0 | No leaks in that section |
| 1 | Micro leaks in that section |
| 2 | Minor leaks in that section |
| 3 | Major leaks in that section |

In order to have a good dataset that allows for the proposed machine learning methodology to work properly, for each of the output possibilities, multiple tests were performed, changing the

duration of the test and pressure applied by the pump or length of the pipes, each of these being also performed multiple times to guarantee that small changes in data are recorded.

These tests allow us to obtain 5607 records containing information from three sensors that were spread through the same path of pipe which analyze the conditions of the pipes in two different zones. Since multiple situations and scenarios were applied to this data collection, it is possible to assume good reliability on the data collected that will train the algorithms so it can be applied to other scenarios.

Although the collected data show information from multiple sensors, each individual record only has the timestamp and the data collected from the sensor. Therefore, the first thing that we intended to analyze was how the data that is inputted into the algorithm can affect its accuracy. For that, two datasets were created using the same collected data. One is the data collected according to the sequence of sensors in the system, and the other is where the data was clustered to show how the entire path of pipes works. This was done in order to understand which of the datasets presents the best results for the intended scenario and system.

When creating the dataset, besides the timestamp and collected sensor data, other features were added to each record based on calculations and pr-processed data from the sensor values. Table 4 presents the fields of each dataset.

**Table 4.** Dataset features.

| Feature | Description |
|---|---|
| id | ID of the data input |
| $sensorID_i$ | Corresponding sensor |
| $value_i$ | Value collected |
| $average_i$ | Average from last 5 values for that sensor |
| $diff\_ref_i$ | Difference from Sensor 1 |
| $diff\_sen_i$ | Difference from previous sensor |
| $hasProblems_i$ | Leaks on that section |

The difference between datasets is that the standard one only has one entry for each of the features while the clustered one has three entries for sensorID, value, average, diff_ref, and diff_sen, each one regarding the three sensors used in the test.

With this, the two datasets created have 5607 and 1869 entries for the standard and clustered sets, respectively.

In order to reach the goal of detecting water leaks and their location, the algorithms must be trained and tested with both datasets. This guarantees that, when used in a real-life scenario, the results will be precise. For that, the datasets were divided into training and testing set, with 70% and 30%, respectively.

*4.2. Model Analysis*

For model analysis, an array of ML algorithms was used in order to determine which is the best one to use in our system.

For our scenario, a total of 12 tests for each algorithm was performed, each test with different parameters, with the objective to train the algorithms to understand which presents the best accuracy so that it could be implemented in our system. Table 5 shows all the performed tests, identifying the used dataset, target, and unused features.

**Table 5.** Test scenarios.

| Dataset | Test N° | Target | Unused Features |
|---|---|---|---|
| Clustered | 1 | hasProblems2 | average, diff_sens, diff_ref |
| | 2 | hasProblems3 | |
| | 3 | hasProblems2 | diff_sens, diff_ref |
| | 4 | hasProblems3 | |
| | 5 | hasProblems2 | diff_ref |
| | 6 | hasProblems3 | |
| | 7 | hasProblems2 | - |
| | 8 | hasProblems3 | |
| Standard | 9 | hasProblems | average, diff_sens, diff_ref |
| | 10 | | diff_sens, diff_ref |
| | 11 | | diff_ref |
| | 12 | | - |

In these test cases, the target output is related to whether a specific sensor has problems or whether everything is working correctly in the system, with "hasProblems2" or "hasProblems3" for the clustered dataset indicating a leak located in Section 2 or Section 3 of the pipe, respectfully, and "hasProblems" for the standard dataset indicating a leak located on that specific section.

Each of the tests was conducted using Python, the scikit-learn library [37], and the Jupyter platform. For each of the algorithms, a script was developed using the corresponding library for classification from scikit-learn and the default configurations were used. As said, 70% of the dataset was used for training and 30% was used for testing. Table 6 presents the results for each test, and Figure 8 represents the same results, allowing for a better analysis.

**Table 6.** Tests results.

| Algorithm | Accuracy [%] | | | | | | | | | | | |
|---|---|---|---|---|---|---|---|---|---|---|---|---|
| | 1 | 2 | 3 | 4 | 5 | 6 | 7 | 8 | 9 | 10 | 11 | 12 |
| RF | 75.40 | 70.05 | 80.75 | 72.73 | 80.75 | 73.62 | 80.57 | 74.69 | 74.51 | 77.00 | 77.82 | 84.79 |
| DT | 76.65 | 70.94 | 80.21 | 71.30 | 80.04 | 72.19 | 79.14 | 70.05 | 74.03 | 77.06 | 77.42 | 84.02 |
| NN | 59.71 | 60.42 | 67.74 | 67.91 | 75.40 | 70.23 | 73.44 | 72.19 | 51.69 | 55.44 | 74.98 | 80.68 |
| SVM | 26.91 | 26.56 | 46.17 | 46.60 | 51.34 | 31.02 | 57.22 | 41.35 | 53.77 | 51.75 | 29.47 | 27.09 |
| XGBoost | 76.65 | 70.94 | 80.93 | 75.22 | 81.46 | 75.76 | 81.11 | 75.04 | 74.99 | 77.01 | 77.78 | 84.73 |

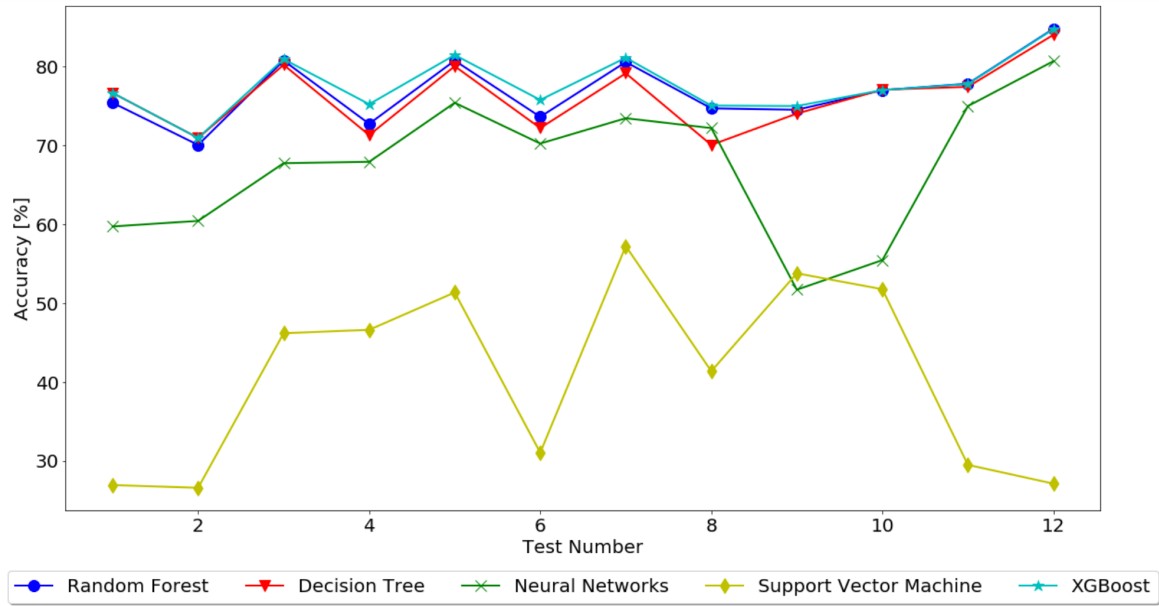

**Figure 8.** Test results.

### 4.3. Remarks

The objective of all of these tests was to determine which had the best accuracy when given inputs related to our water leakage system as well as to check which was the best dataset composition. With the results from the performed tests, it is possible to make conclusions and to define the best algorithm and dataset for our intended system.

It is possible to attest that the datasets have similar results but with different behaviors, with the accuracy for most of the algorithms being within the same range, about 70% to 85%, excluding SVM, which for both datasets presented an accuracy within the range of 27% to 53%.

We can also perceive that the variables have an important role for the output of each test, always having an influence on the algorithm that is tested. In the clustered dataset, for the algorithms RF and DT, we can see that, from the first to the second tests using the same target (tests 1 and 3 for "hasProblems2" and tests 2 and 4 for "hasProblems3"), the tendency is for the accuracy to increase with less unused features. This behavior then changes (tests 3 to 5 and 7 for "hasProblems2" and tests 4 to 6 and 8 for "hasProblems3") as the accuracy decreases, concluding that these algorithms work better when the number of unused features is bigger. For the algorithms NN, SVM, and XGBoost, the behavior is the same for all targets when the unused features decrease, with all the algorithms increasing in accuracy from test to test, thus concluding that these algorithms work better with less unused features. For the standard dataset, all tests have an accuracy increasing in each test with lesser unused features, with the exception being SVM, where the accuracy decreases when the number of unused features also decreases.

As we can see from the results, SVM showed the worst accuracy in almost all tests, with most results being below 50% accuracy, so it can be discarded as the algorithm to used. The NN algorithm, even though better than SVM with no outputs lower than 50% accuracy, still got final results less than expected and intended. RF, DT, and XGBoost presented very similar final outputs in terms of accuracy, with no results below 60% accuracy and with a few final outputs above 80%. It is also important to mention that, for RF, DT and XGBoost, the accuracy increases depending on fewer parameters being dropped from the dataset.

With the presented results, is possible to conclude that, when using the data as it is collected, without clustering, the accuracy increases, with the best dataset being the standard one. As for the best algorithm, RF shows the best accuracy, with approximately 85%, when using all the features on the dataset.

It also important to mention that the obtained results, although lower than the ones presented in related works as presented in Section 2, fall in line with the work presented by those papers. Compared to [15], which obtained 92% accuracy for leak detection using SVM, our SVM approach did not get even close, with only a 57% accuracy in the best-case scenario. As said before, that study only used a 200-sample dataset to train their model, which can be why they got almost 40% improvement. When referring to the work in [16], we also conclude that RF is the best algorithm for leak detection. Our results are similar across all algorithms, which helps validate our approach and test.

## 5. Experimental Implementation

Using the results from Section 4 associated with the system architecture presented in Section 2, the final system was assembled, including the trained random forest model for data analysis in real time, as sensor values reach the cloud server.

The goal of the proposed developed system was to be fitted on a set of pipelines that supply water in irrigation systems or households to warn the user when situations such as leaks start to appear and their location, not only to notify the user but also to help prevent this type of situation from evolving to bigger ruptures or other problems, such as water and monetary waste.

Due to the current pandemic situation, the planned real-case implementation was not possible, and as such, an experimental implementation was conducted, simulating the real environment.

For that, the same scenario from the machine learning training test was used, simulating a pipeline that supplies water in an irrigation system, using 3 sensor nodes, and evaluating two pipeline zones, as shown in Figure 7. As in the previous data collection test, the system was left running for several hours, collecting data from individual sensors as water flows through them and sending those values to the server to be analyzed and to predict whether leaks are occurring and where they are.

The objective of this experimental implementation was to understand if the developed and trained system can perform under new situations and outside the training data. As such, the test started with a new set of pipes in perfect conditions and, over time, holes were made in them to simulate leaks and to check if the system can detect them and warn the user about their location in real time. To check the accuracy of the system, as each hole was made, the timestamp and location were recorded to later compare with the results from the system.

### *Deployment and Maintenance Costs*

Considering the presented solution, the main goal was to develop a low-cost way to deploy a network of nodes capable of monitoring irrigation pipelines. For that, each kilometer of pipeline should have at least 10 sensor nodes, one at each 100 m, to narrow detection of the precise location of the leak, with the cost of each sensor node being around 20 €. Besides that, one aggregation node can cover up to 2000 nodes in an 2 km line of sight range, with the cost of 30 €, and a monthly data plan cost of about 0.20–0.80 € per sensor node, depending on the irrigation period.

Considering the market solutions presented in Section 2, the [9] solution cost about 2500 € for a portable device that needs to be used manually and the [11] solution, that can be installed in two pipes using only two sensors per pipe, costs about 5000 €. When compared to the academic solution stated, the costs are similar, depending mainly on the type of sensor used and the number of nodes.

## 6. Results

After implementation of the final system, the system prototype was put to test, working for several hours during a day, collecting a total of 3703 data entries from three different sensors. The actual test, in a real environment, would work throughout a longer period of hours and days since irrigation in fields is done at specific times of each day, but as mentioned before, this was not possible, so a smaller system was used.

As said, the developed system is able to analyze the data as soon as it was gathered by the sensor nodes and transmitted to the server. The RF trained model included in the Python script that receives

the messages and stores the values in the database is able to evaluate in real time whether a leak is present as well as the location based on sensor data. Figure 9 presents the collected sensor data from the entire experimental study as well as the predicted value for the existence of leaks in each of the sections under study.

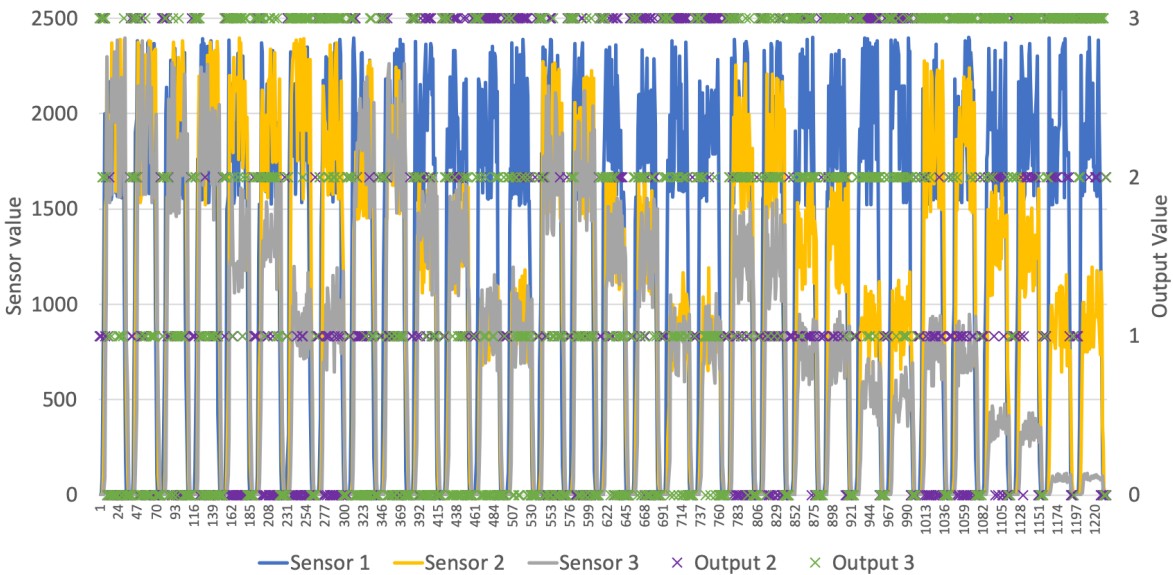

**Figure 9.** Test results.

Sensors 1 to 3 show values collected from the different nodes, with the value frequency of the water rotor. Output 2 refers to a problem located in the first section of the pipeline, between sensors 1 and 2, and output 3 refers to a problem in the next section, between sensors 2 and 3, according to the output presented in Table 3.

As said before, to evaluate the accuracy of detection of leaks and their locations, when holes were made in the pipes, the timestamp was annotated to allow a comparison between the real and the predicted scenario. Figure 10 shows the confusion matrix from the obtained results, allowing to see when true positives, true negatives, false positives, and false negatives were obtained for each of the possible outputs. With this, it is possible to understand the situations where the outputs were wrongly predicted and to calculate the accuracy of the system for that particular test.

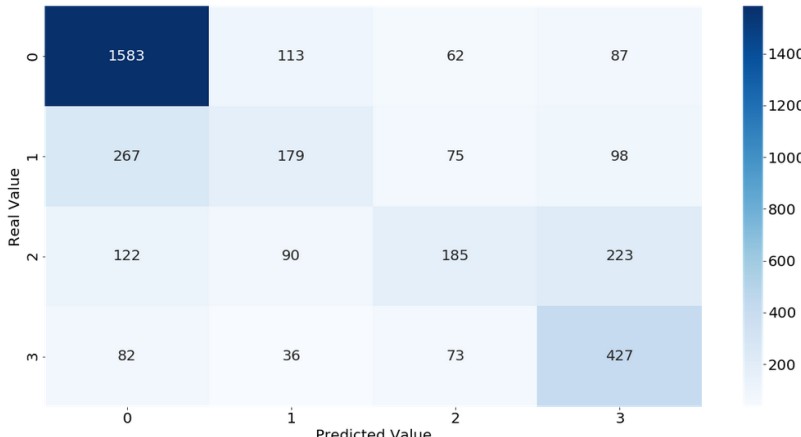

**Figure 10.** Results matrix analysis.

## 7. Discussion

The first thing to notice from Figure 9 is that the developed system is able to detect fluctuations of the water flow alongside the pipeline, as values increasem maintain a certain range when water flows, and goes to 0 when water is not flowing. Another thing to notice is that the system nodes are able to function apart from each other, as can be seen when different flows are detected in different sections of the pipeline. Finally, it is possible to check that. for every sample collected. an output was generated, meaning that the value was successfully collected, transmitted, and analyzed by the system. With this, it is possible to conclude that the developed system works as intended and is ready to be implemented in a real-case scenario, being one of the major contributions from this paper.

As for the leak detection system, it is possible to see that the predicted outputs of the system are able to detect not only if a leak exists but also the section where it occurs based on the data collected from the sensors. As it is possible to see in Figure 9, outputs 2 and 3 identify the problems in the corresponding pipeline sections.

In terms of accuracy when predicting leaks, Figure 10 shows a confusion matrix with the results obtained from the RF classification algorithm, showing what really happen in the system and what the system detected. Looking at the diagonal in the matrix that presents the true positives, where the values were correctly predicted, it shows that the system is able to detect more correct situations than wrong ones, with 2374. There are some mistakes between the 0 and 1 output values, indicating that, when detecting minor leaks, the system still needs to be improved. Also, some situations where the output was 2 were identified as 3. The more concerning situations are major leaks that were identified as non-problems in 82 cases and situations where no leak was present but was identified 87 times as majors' leaks. This showcases that the accuracy of the system is still not perfect but is in line with the results from the training, where the RF model got only 85% accuracy.

Overall, the system obtained a 75% accuracy when detecting leaks in the experimental system, which is a good result. Although it is 10% lower in terms of accuracy when compared to the trained algorithm, that can be justified by the small dataset used to train the model that can lack some of the specific solutions that the system can encounter or some environment specification that differs between tests.

Besides this difference, in conclusion, a 75% accuracy is still a great result than can help improve the early detection of leaks, proving that the developed system and trained models are great contributions to solving the undetected leak problem.

## 8. Conclusions

In this paper, an IoT system was presented, capable of monitoring water distribution systems and of finding and locating with precision water leaks by using low-cost sensors and by collecting data in real time. The main goals of the system were achieved, and the system proved to be efficient and reliable. Every part of the system was previously and properly tested before reaching the final prototype and its implementation.

Trough the ML test scenarios, it was possible to study which is the best algorithm to use in this scenario. Multiple models were tested in various configurations and different datasets in order to achieve the best configuration possible. It was possible to conclude not only that, when using more features, the accuracy increases but also that random forest achieves the best accuracy in almost every scenario, making it the best classification algorithm to use, with almost 85% accuracy.

Besides the theoretical implementation and machine learning study, an experimental implementation was also presented to validate the methodology used and the results obtained in our study in a real case scenario.

In that, the system was able to detect, with a 75% accuracy, the presence of leaks in a pipeline. Although the results obtained are lower than those obtained in the laboratory test phase, it is possible to conclude that our system is able to help prevent water lose and pipe malfunction. The lower results indicate that the system still needs some improvements and further testing.

As a final note, it is possible to verify that the developed system meets all the conditions essential for a complete and functional intelligent system and that the system is at the level of the other systems already in the market and in the academic world, with the benefit of being a low-cost version but with high quality, efficiency, and reliability, which means that, in the future, it can be adjusted and introduced as a new market solution.

For future work, as already said, the system needs some improvement, mainly to improve the accuracy of the RF model when detecting minor leaks. Also, implementation on the proposed scenario in Section 5 in a real irrigation pipeline system will be done to further evaluate the system when the pandemic situation is over.

**Author Contributions:** conceptualization, J.A.C., A.G. and P.S.; methodology, A.G.; software, J.A.C. and A.G.; validation, A.G.; formal analysis, A.G.; investigation, J.A.C. and A.G.; resources, A.G.; data curation, A.G.; writing—original draft preparation, J.A.C. and A.G.; writing—review and editing, P.S.; visualization, A.G.; supervision, P.S.; project administration, A.G.; funding acquisition, A.G. and P.S. All authors have read and agreed to the published version of the manuscript.

**Funding:** This work was supported in part by ISCTE—Instituto Universitário de Lisboa from Portugal under the project ISCTE-IUL-ISTA-BM-2018.

**Conflicts of Interest:** The authors declare no conflict of interest.

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
