# Peer review of "Precise Water Leak Detection Using Machine Learning and Real-Time Sensor Data"

_2624-831X, doi:10.3390/iot1020026_

Round 1
Reviewer 1 Report
This paper is about the design of a water leaks detection system for monitoring water distribution systems. The manuscript outlines the serveral pieces required for implementing real-time water leaks detection and it evaluates several machine learning methods for detecting water leakages, providing a valuable benchmark for future implementations.
The readability of the paper must be improved. In general, the writing is not linear and hard to follow for readers. The overall organization is quite good, however, the authors can improve it by further discussing some important concepts. The paper seems to be a solid work from an engineering perspective, but the current status of writing penalizes its quality.
Some comments:
The abstract needs rewriting. First, the authors should provide an introduction to the field of water detection systems (2 lines). Then, the authors can outline what their paper proposes.
The Introduction could greatly benefit from rewriting. At line 22 the authors can cite some related work in the field of IoT-based water leak detection systems.
At line 37 the authors claim that the proposed system can identify the location of water leaks. Is the system capable of finding leaks' locations?
If yes, this concept should be further discussed within the paper.
It is worth citing some more works in the Related section.
Figure 3 should be improved! It should better highlight the interactions between components. I believe that representing sensors on the left and users on the right is more accurate.
In which way the server interact with the WSN nodes (right arrow in Figure 3)?
The authors can enhance the description of aggregation and sensor nodes by describing how they are powered. Is the aggregation node attached to the electrical grid? Does it require an external battery pack?
What about the sensor nodes? Do they have an average lifetime?
Experiments exploit a small-scale testbed the authors developed to create their dataset for evaluating different machine learning algorithms. This Section demonstrates the efforts the author did in evaluating different machine learning algorithms for identifying water leaks in a water distribution system. Here, the writing can be improved to better highlight the authors' contributions.
Table 5 is very hard to understand. The authors should spend more efforts in improving the readability of the Table and its description. This is a major flaw in the paper
Reviewer 2 Report
This paper presents a system based on a wireless sensor network, designed to monitor water distribution systems and detect water leaks.
The system includes a Machine learning algorithm to analyze the data collected and detect the water leaks.
Although there are other systems already in the market and academic world, authors claim that this new system is a low-cost version but with high quality, efficiency and reliability.
The subject of the paper is interesting and actual and it has a considerable set of references.
The paper is well structured, but the writing needs a major reviewing process because it has many spelling errors and some of the sentences are not intelligible.
In my opinion authors should be encouraged to review the writing and resubmit it for publication.
Author Response
Point 1: The paper is well structured, but the writing needs a major reviewing process because it has many spelling errors and some of the sentences are not intelligible.
In my opinion authors should be encouraged to review the writing and resubmit it for publication.
Response 1:
The entire paper was revised to improve the quality and writing. Several typos were corrected, sentences were written in a more fluid and comprehensive way, and some concepts were added.
Reviewer 3 Report
This paper presents an approach to implementing an IoT-based system for detecting water leakages, utilizing an ML approach. The subject of this work is interesting and relevant to the audience of this journal.
The abstract is a bit short and should be revised to provide a more complete picture of the paper and the problem discussed here.
The introduction is also a bit short, leaving out several aspects that are typically included here, e.g., no mention of the different types of systems used for water leakage detection, technologies currently or previously utilized, a more detailed description of the contributions of the paper and the results discussed, the structure of this paper.
Furthermore, the related work section in my opinion is underdeveloped. The authors should add additional references and description for existing work in this application domain - e.g., there are a number of papers in MDPI Sensors or Applied Sciences journals for LoRa-based water quality monitoring and remote metering, along with leakage detection. There are also products in the market currently available - with most probably not using ML, but still some mention should be included. The authors should also use the references included in this section to explain where their approach differs from the existing state-of-the-art and in which ways they improve upon previous results, with a more extensive discussion (currently only one phrase is included about this aspect).
Moving on to the rest of the paper, the scale and scope of the experiments described is a bit limited. The scale of the prototype utilized is very small. Moreover, I think that the authors should go back and implement the experiments as they had originally envisioned them (as mentioned in Section 5). Just by including experiments with a larger scale of water monitoring system would add more credibility to the approach and results presented in the paper. I think the current scope and scale of the experimental system does not meet the requirements for a publication. I would also suggest that the authors provide examples with respect to how an ML-based approach would fare against other approaches, e.g., in terms of financial costs for infrastructure, operation, etc. This is implied as a benefit in the paper, but is not discussed in detail.
Regarding the presentation of the paper, in general the text is easy to follow and I appreciate the effort put by the authors in the figures included. However, some figures are of low resolution (e.g., Fig.1), while there are many small errors and typos throughout the text. Some examples in the first few pragraphs:
- "important to also mentioned"
- "is maximize"
- "to complex"
- "comes up the proposed"
- "that have prejudice"
etc. There are paragraphs with just one short sentence, e.g. "Figure 5 is represented the sensor node."
Fig. 2 could be enhanced by including some details on the technologies, e.g. LoRa for wireless networking.
Overall, I think the authors should make many changes to their text, and I highly encourage them to revise their paper. There is potential in this work, but at its current state it should be rejected.
Round 2
Reviewer 1 Report
I thank the authors for addressing all my comments.
I believe that the current version of the manuscript is of improved quality. I appreciate the authors' efforts done to enhance the quality of the whole paper.
Before publication, please take care or do another minor revision to the paper.
I noted that there are some missing references.
I do not have further concerns or questions.
Author Response
The pointed revisions were done:
- Add the missing references
- Review the paper for typos
Reviewer 3 Report
Since this is a revision of the original paper, the authors have improved several of the original paper's shortcomings and overall the paper is better. The authors have made significant efforts to improve sections such as the introduction and related work that were very lacking in the original version of the paper.
A number of errors and typos have been corrected in this revision of the paper, and the figures have been revised as suggested by the reviewers (e.g., Figure 2). The paper has also overall increased significantly in size (from 15 to 18 pages).
As regards some more minor suggestions, I would suggest that the authors add to the abstract some mention of the ML approaches compared, or their number, to give an indication to the readers regarding this aspect. I would also suggest that the authors use the term "LPWAN" instead of "LR-WAN".
Although the paper has been improved, I continue to not be convinced by the scale of the experiments included in the revision, and would strongly suggest that the authors conduct additional experiments to back up their work in this paper.
However, the authors can also try to strengthen the contribution of this paper via other ways.
"Low cost" is mentioned 6 times in the paper, but there is no actual explanation of what this entails. I would suggest that the authors provide a more clear description of the costs involved and also provide a comparison of their approach vs. other more conventional approaches. What are the benefits? Are they using less nodes, etc.? What would be the deployment costs if they had implemented the experiments as originally planned? The authors could add e.g. a new subsection addressing these aspects.
Moreover, the authors could expand the description of their battery lifetime description. Also, in this section, please correct "AS such" and "autosustainable" should be replaced with "autonomous".
Furthermore, the authors could expand a bit their description for communication requirements (bandwidth, range, etc.), e.g., through an indicative scenario. They could also explain why LoRaWAN could not satisfy the application requirements and why they instead chose NB-IoT with the associated costs (they could also give an indication of these costs). I think also explaining a bit better why they used LoRa and not LoRaWAN for the nodes could clarify some additional aspects to the readers.
Finally, I suggest that the authors correct the remaining errors and typos in the text - it has been significantly improved, but there are some remaining in the current version of the text.
Based on the above comments, I propose that the authors submit a major revision of their paper.
